# Current and Novel Therapeutic Approaches for Treatment of Diabetic Macular Edema

**DOI:** 10.3390/cells11121950

**Published:** 2022-06-17

**Authors:** Muhammad Z. Chauhan, Peyton A. Rather, Sajida M. Samarah, Abdelrahman M. Elhusseiny, Ahmed B. Sallam

**Affiliations:** 1Department of Ophthalmology, Harvey and Bernice Jones Eye Institute, University of Arkansas for Medical Sciences, Little Rock, AR 72205, USA; mzchauhan@uams.edu (M.Z.C.); parather@uams.edu (P.A.R.); schauhan@uams.edu (S.M.S.); amelhusseiny@uams.edu (A.M.E.); 2Miami Integrative Metabolomics Research Center, Bascom Palmer Eye Institute, University of Miami, Miami, FL 33136, USA

**Keywords:** diabetic macular edema, therapeutics, VEGF, laser photocoagulation, intravitreal injection, novel pharmacotherapy

## Abstract

Diabetic macular edema (DME) is a major ocular complication of diabetes mellitus (DM), leading to significant visual impairment. DME’s pathogenesis is multifactorial. Focal edema tends to occur when primary metabolic abnormalities lead to a persistent hyperglycemic state, causing the development of microaneurysms, often with extravascular lipoprotein in a circinate pattern around the focal leakage. On the other hand, diffusion edema is due to a generalized breakdown of the inner blood–retinal barrier, leading to profuse early leakage from the entire capillary bed of the posterior pole with the subsequent extravasation of fluid into the extracellular space. The pathogenesis of DME occurs through the interaction of multiple molecular mediators, including the overexpression of several growth factors, including vascular endothelial growth factor (VEGF), insulin-like growth factor-1, angiopoietin-1, and -2, stromal-derived factor-1, fibroblast growth factor-2, and tumor necrosis factor. Synergistically, these growth factors mediate angiogenesis, protease production, endothelial cell proliferation, and migration. Treatment for DME generally involves primary management of DM, laser photocoagulation, and pharmacotherapeutics targeting mediators, namely, the anti-VEGF pathway. The emergence of anti-VEGF therapies has resulted in significant clinical improvements compared to laser therapy alone. However, multiple factors influencing the visual outcome after anti-VEGF treatment and the presence of anti-VEGF non-responders have necessitated the development of new pharmacotherapies. In this review, we explore the pathophysiology of DME and current management strategies. In addition, we provide a comprehensive analysis of emerging therapeutic approaches to the treatment of DME.

## 1. Introduction

Diabetes mellitus (DM) is a metabolic disorder characterized by persistent hyperglycemia due to insulin secretion or action malfunction. DM is classified as Type 1 or Type 2: Type 1 DM is due to an autoimmune reaction that leads to the loss of insulin-producing cells in the pancreas, resulting in absolute insulin deficiency, while Type 2 DM is associated with insulin resistance and impaired secretion, resulting in relative insulin deficiency. DM is the leading cause of end-stage renal disease, non-trauma-related amputations of the lower extremity, and, most relevant for this review, adult-onset blindness in the United States (US) [1]. On a global scale, DM has been listed as one of the main contributors to vision loss and the only cause of blindness, showing a global increasing trend in age-standardized prevalence between the years 1990 and 2020 [2]. DM affects multiple intraocular structures in the anterior and posterior segments of the eye, as well as extraocular structures. On the ocular surface, elevated glucose levels are toxic to meibomian gland epithelium, causing abnormal tear film production and resulting in dry-eye syndrome [3]. In the cornea, DM has been shown to lead to epithelial layer fragility, abnormal collagen bundles in the stroma, and decreased cell density in the endothelium [4]. In the anterior chamber angle, the growth of new blood vessels causes a secondary elevation in intraocular pressure, leading to neovascular glaucoma [5]. In the lens, high glucose levels lead to the conversion to sorbitol by aldose reductase, causing osmotic stress and the eventual development of cataracts [6]. In the optic nerve, vascular leakage and axonal edema result in swelling of the optic disc and the development of diabetic optic neuropathy [7]. Hyperglycemia-induced damage to nerve cells is the most common etiologic subset of oculomotor nerve palsy [8]. In the orbit, uncontrolled DM has the potential to impact immune cell chemotaxis and phagocytic efficiency, permitting the growth of Mucorales species and the ultimate development of mucormycosis, which can extend to the brain and cause severe damage [9]. However, the single leading cause of visual dysfunction in the diabetic population is clinically significant diabetic macular edema (DME) [10], followed by diabetic vitreous hemorrhage and tractional retinal detachment.

The Wisconsin Epidemiologic Study of Diabetic Retinopathy found the incidence rates of DME to be 20% in patients with Type 1 DM and between 14 and 25% in patients with Type 2 DM over 10 years [11]. Nevertheless, DME may present at any stage of DR. The prompt management of DME is crucial to prevent irreversible damage to photoreceptors. Historically, laser retinal photocoagulation has been the standard of care for DME patients [12]. However, with the advent of pharmacotherapies, such as anti-vascular endothelial growth factor (VEGF) and corticosteroids, there has been a move away from the use of lasers. The growing prevalence of DME and, as a result, the increase in the number of patients who are not benefiting from anti-VEGF and corticosteroid therapy, has provided an impetus to shift the emphasis to novel therapeutic targets and approaches [13].

In this review, we provide an overview of the pathogenesis of DME and discuss the current management approaches. In addition, we offer a thorough review of novel therapeutic approaches to DME treatment. 

## 2. Overview of the Pathogenesis of Diabetic Macular Edema

To better understand the current and new therapeutics for DME treatment, it is imperative to grasp the mechanisms of development and critical clinical presentations. This section provides a brief overview of the pathophysiology and clinical/pathological features of DME. Persistent hyperglycemia associated with a DM of any type is the leading cause of chronic complications and pathology. However, the risks of developing DME differ between patients with Type 1 DM and Type 2 DM [14]. Blood glucose is typically maintained within narrow ranges (typically 70–120 mg/dL) through intricate endocrinological and cellular functions in various organs and tissues. In DM, glucotoxicity causes chronic complications through multiple pathways, namely, through the formation of advanced glycation end products (AGEs), activation of protein kinase c, disturbances in polyol pathways, and overactivation of the hexosamine pathway [15]. 

The development of AGEs, which are molecules (i.e., proteins, lipids, etc.) that become glycosylated due to exposure to sugars, modify the molecular structure, resulting in a decrease in or loss of function of enzymes and vital cellular components [16]. Hyperglycemia also upregulates intracellular levels of diacylglycerol (DAG), which functions as a second messenger signaling lipid. DAG acts as a physiological activator of protein kinase C (PKC) [17]. The activation of PKC results in the upregulation of VEGF [18,19] and alters vascular permeability, resulting in hypoxia [20] and blood–retinal barrier (BRB) impairment [21]. The polyol pathway is a metabolic pathway where glucose is converted to fructose. In this pathway, glucose is first converted to sorbitol by the aldose reductase, oxidizing NAPDH to NADP+ in the process. In the setting of chronically elevated blood glucose, there is increased flux through the polyol pathway, and NADPH is consumed at high rates. The excessive consumption of NADPH reduces glutathione regeneration, causing elevations in reactive oxygen species that play a role in the structural abnormalities seen in retinal microangiopathy [22]. In the hexosamine biosynthesis pathway, fructose-6-phosphate is converted to UDP-N-acetylglucosamine, providing building blocks for N- and O-linked glycosylation. The increased shunting of glucose through the hexosamine pathway has been shown to promote insulin resistance as well as injury and inflammation by inducing the expression of nuclear factor κB (NF-κB)-dependent genes [23]. More recently, the upregulation of arginase, an enzyme that catalyzes L-ornithine and urea from L-arginine, has been linked to oxidative stress and peripheral vascular dysfunction in diabetes [24,25,26]. Another recent work utilizing a quadra-omics approach (proteomics, lipidomics, metabolomics, and transcriptomics) examined blood samples from patients with Type 1 DM and discovered the activation of TGF-β, VEGF, NF-κB, and arginase with the inhibition of miRNA Let-7a-5p [27]. These pathways and mediators are thought to incite oxidative stress by overproducing reactive oxygen species, causing defects in angiogenesis, and activating pro-inflammatory pathways.

These overarching biochemical mechanisms are thought to play a role in the development of DR through neurodegenerative and vascular changes in multiple cellular structures. DR is unlikely to be due to one single pathogenic mechanism or disturbance in one cell type. However, most relevant to the development of DME, which may occur at any point in the pathogenesis of DR, are alterations in vascular endothelial cells, tight junctions, pericytes, and basement membrane, ultimately leading to tissue ischemia, vascular leakage, and disorganization of the BRB (Figure 1A). Clinically, this manifests in fundoscopic imaging as capillary microaneurysms, neovascularization, blot hemorrhages, hard exudates, and cotton wool spots (Figure 1B). These disturbances result in the leakage of fluid, proteins, and lipids from retinal vessels (i.e., vasogenic), with a concomitant swelling of Müller cells (i.e., cytotoxic), resulting in DME through a complex series of events (Figure 2A,B) [28]. Chronic swelling of the macula leads to damage to the neural retina and subsequent loss of photoreceptors (Figure 2C). DME can cause irreversible retinal disorganization and subsequent loss of central vision without prompt treatment.

## 3. Current Therapeutics

### 3.1. Primary Glycemic Control

The maintenance of blood sugar levels, among other modifiable risk factors (hypertension, hyperlipidemia, etc.), is widely considered to be one of the most critical factors known to decrease the risk of DR progression [30]. Findings from three consequential clinical trials have shown decreases in the incidence of and progression to DR in the setting of glycemic control. The Diabetes Control and Complications Trial (DCCT) was a controlled clinical trial conducted between 1982 and 1993 on 1441 subjects with T1DM [31]. This clinical trial aimed to determine the benefits of intensive treatment in T1DM. Patients enrolled in the trial were randomized to an “intensive treatment” or “conventional treatment” group. Intensive treatment was defined as aiming to reach a glycemic control close to the nondiabetic range through daily insulin injections or insulin pump therapy, while conventional treatment aimed to maintain a safe asymptomatic glucose control [32]. The intensive treatment group, which held an average glycated hemoglobin (A1C) (HbA1c) of 7.2%, was found to lead to a significant reduction in the incidence and progression of DR (54% and 76%, respectively) compared to the conventional treatment group (HbA1c of 9.2%). Patients in the intensive treatment group maintained this risk reduction for at least four years after the end of the trial [33]. One adverse finding was that if a patient with uncontrolled DM rapidly corrected hyperglycemia, there was an observed increased risk of DR aggravation [34]. These negative observations were found to be temporary, with the long-term data in support of intensive glycemic control. 

Another clinical trial was the United Kingdom Prospective Diabetes Study (UKPDS) that started in 1977, which aimed to determine whether patients with T2DM who had intensive glycemic control showed a reduced risk of macrovascular or microvascular complications [35]. Findings from the UKPDS revealed that T2DM patients who underwent intensive glycemic control maintained an average HbA1c of 7.0% and showed a 21% reduction in the progression of DR compared to those with conventional glycemic control [36]. In addition, the Chennai Urban Rural Epidemiology Study (CURES) found that a stepwise increase in HbA1 levels resulted in a significant increase in the prevalence of DR [37]. Findings from the Action to Control Cardiovascular Risk in Diabetes (ACCORD) Trial published in 2010 found that patients with T2DM who maintained HbA1c levels below 6.0% had a significantly lower rate of DR progression than HbA1c levels targeted at 7.0% [38]. However, the ACCORD Trial also found that once a patient has progressed to developing DME, controlling blood sugar, hyperlipidemia, and blood pressure had no benefit in modifying the prognosis [39]. The results from these landmark clinical trials suggest that patients with T1DM or T2DM who maintain an average HbA1c value of at least ≤7.0% have a decreased risk of incidence and progression of DR. Overall, tight glycemic control is most beneficial when started early in the DM pathogenesis, before progression to DME. However, despite substantial efforts to control hyperglycemia in the diabetic population, approximately 38.2% of patients with diabetes for 20 years will develop retinopathy of some kind, necessitating the use of ophthalmic-specific treatments [40]. 

### 3.2. Laser Photocoagulation

Before the introduction of intravitreal anti-VEGF agents, laser photocoagulation was the standard of care for DME treatment. It was introduced to the field of ophthalmology over 25 years ago [29,41]. In 1985, in the Early Treatment Diabetic Retinopathy Study (ETDRS), a multicenter randomized trial exploring the benefits of laser therapy for retinopathy, it was demonstrated that photocoagulation could halve the risk of moderate visual loss over three years from 24% to 12% compared to deferred treatment [42]. Two laser treatment approaches were described in the EDTRS: Focal and Grid.

A focal laser was applied to focal microaneurysms, intraretinal microvascular abnormalities, and small capillary leakages of between 500 and 3000 microns from the fovea. In contrast, a grid laser was performed in areas of diffuse leakage with macular thickening with or without non-perfusion [43]. In the EDTRS, microaneurysms contributed to most focally treated lesions. Grid treatment was not recommended within 500 μm of the center of the macula or 500 μm of the disc margin to preserve central vision. The mechanisms of action of focal/grid lasers have not been fully elucidated. The purported mechanism of action of laser photocoagulation is mainly the closure of the leaking microaneurysm [44]. Some studies have shown that laser induces biochemical effects, including the decreased production of VEGF due to decreased retinal oxygen demand and increased phagocytosis by glial cells [45,46]. 

Despite the advantages of the macular laser shown in the ETDRS, less than 3% of patients with initially good visual acuity (VA) (20/40 or better) who underwent laser treatment improved VA by three lines or more (15 letters) from baseline [42]. Macular photocoagulation is also not suitable for foveal involving DME and in eyes with significant foveal avascular zone disruption. In more recent years, the use of laser photocoagulation was relegated to DME without central involvement. A randomized clinical trial by Baker et al. [47] examined vision loss after 2 years among eyes treated with either aflibercept, laser photocoagulation, or observation. They found that among 702 participants at two years with a center involving DME with a VA of ≥20/25, there was no significant difference in vision loss when eyes were initially managed with aflibercept (16% (33/205)), laser photocoagulation (17% (36/212)), or observation (19% (39/208)), indicating that no treatment may be beneficial if VA is stable (*p* = 0.79). Therefore, evidence suggests that there is no need to treat these eyes. The potential side effects of macular laser are principally due to the thermal effects of photocoagulation and include choroidal neovascularization, subretinal fibrosis, and visual field loss [48]. Laser performed near the central fovea may result in a drop in VA after laser treatment. The burns induced by laser therapy may also be associated with para-central scotomas. It has been proposed that using lighter and less intense laser burns than what was originally specified in the ETDRS may reduce side effects. In 2005, a prospective randomized clinical trial of diabetic patients with non-proliferative DR (NPDR) and DME were randomized to classic or barely visible laser treatment [49]. In the “Light” laser treatment group, the energy used was just enough to cause barely visible burns in the RPE. At one year, no statistically significant difference in edema reduction, visual improvement, visual loss, change in contrast sensitivity or mean deviation in the central 10 degrees was observed between groups (*p* > 0.05), suggesting that “light” photocoagulation for clinically significant DME is likely to be as effective as conventional laser treatment. 

Photocoagulation uses conventional continuous-wave laser systems that damage the neural retina through the dispersion of thermal energy from the RPE. In comparison, waves that use short pulses, also called micropulses, cause less thermal damage in comparison to traditional continuous wave treatment, as retinal tissue is allowed to cool between pulses [48]. Subthreshold laser, also known as the “invisible” laser, refers to photocoagulation that does not produce visible intraretinal damage or scarring after treatment through ophthalmic imaging methods such as biomicroscopy, and spectral-domain optical coherence tomography, or fundus autofluorescence. The subthreshold micropulse laser is thought to have little effect on outer retinal tissue and is primarily absorbed by the RPE melanin. Although the exact mechanisms of the micropulse laser are not completely understood, it is hypothesized that it works through photostimulation of the retina pigment epithelium pump and thereby enhances intraretinal liquid resorption. Subthreshold micropulse laser irradiation has been shown to trigger substantial elevations of aquaporin 3 gene expression and a subsequent increase in the drainage of subretinal fluid [50]. In addition, it has been shown to decrease aqueous humor pro-inflammatory molecules [45]. Lavinsky et al. [51] compared a modified ETDRS focal/grid laser photocoagulation protocol with normal-density or high-density subthreshold micropulse photocoagulation in patients with DME. After one year, the high-density group showed the greatest best-corrected improvements in VA (0.25 logMAR; Snellen equivalent ≈ 20/36) compared to patients who underwent a modified ETDRS approach (0.08 logMAR; Snellen equivalent ≈ 20/24) (*p* = 0.009). Another advantage of micropulse photocoagulation is that it is relatively safer for centers involving DME compared to focal/grid photocoagulation [52]. However, a recent study examining the visual and anatomical outcomes of patients with centers involving treatment-naïve or refractory DME treated with micropulse photocoagulation found that macular thickness and VA at three months were unchanged [53]. 

Selective retinal therapy (SRT) works through the application of a series of 30 laser pulses from a 527 nm laser in the range of 450–800 mJ/cm^2^ per pulse [54,55]. Unlike the micropulse laser, SRT laser lesions are identifiable by fluorescein angiography. A prospective interventional uncontrolled pilot study of SRT found a statistically significant improvement in VA at 6 months in patients with clinically significant DME: mean BCVA improved from a baseline of 43.7 to 46.1 letters (*p* = 0.02) [56]. SRT has not been a widely adopted treatment modality for DME, in part due to the lack of visible retinal changes and difficulty defining energy requirements for therapeutic RPE damage [57]. 

The Pattern Scan Laser (PASCAL) Photocoagulator, a frequency-doubled Nd: YAG diode-pumped solid-state laser with a wavelength of 532nm, was announced by Optimedia Corporation in June 2006. For treatment, the operator can choose from a variety of arcs, circular grid patterns and grid sectors, or employ a rectangular array. The main benefit of using the PASCAL for DME treatment is that the initial pulse time of 100 milliseconds can drop to only 10–30 milliseconds and may lead to less pain during the procedure [58]. It has also been demonstrated in several histologic studies that shorter pulse-duration burns cause less tissue damage to the inner retina [59]. 

In addition to the laser procedures discussed above, there are several other novel laser approaches, such as image-guided navigated laser delivery and endpoint management. Navigating laser treatment (NAVILAS) is a recent technology that uses integrated imaging and navigation technologies to perform retinal laser treatment. This technology can deliver both long and short laser pulses in various patterns. The built-in camera system captures true-color, infrared, and fluorescein angiography images with a 50-degree view of the posterior pole. To accommodate for eye movements during treatment, the eye-tracking system stabilizes the position of the targeting beam and laser onto the retina. The NAVILAS system also records the laser spots and patterns sent to the retina, allowing for more thorough treatment and the avoidance of overtreatment. This is especially useful when employing a subthreshold laser, as the laser spots are virtually undetectable during and after therapy. NAVILAS has been demonstrated to be safe and effective in the treatment of DME with accompanying improvements in VA (0.695 to 0.477 (Snellen equivalent ≈ 20/99 to 20/60), *p* < 0.001) and central foveal thickness (248 to 220, *p* < 0.001) up to one year after treatment [60]. In addition, NAVILAS treatment accuracy was found to be superior to conventional laser for DME treatment, with a microaneurysm hit rate of more than 90% versus 72% with conventional laser (*p* < 0.01) [61]. At present, laser treatment is mainly reserved for progressing non-central DME to stabilize vision. 

### 3.3. Anti-Vascular Endothelial Growth Factor

Intravitreal pharmacotherapy based on VEGF inhibition is currently the primary approach utilized for the treatment of DME. Chronic hyperglycemia causes oxidative damage to the vascular endothelial cells and, through a series of events, leads to ischemia. Following ischemia, a variety of growth factors, including insulin-like growth factor-1, fibroblast growth factor-2, tumor necrosis factor, and VEGF, are overexpressed [62]. VEGF mediates several processes involved in the pathogenesis of DME, including angiogenesis, protease production, endothelial cell proliferation, migration, and neovascularization. The vascular endothelial growth factor also relaxes endothelial cell junctions and increases vascular permeability. The process of angiogenesis, or the formation of new vessels, primarily occurs in areas with low oxygen. Low oxygen tension induces VEGF production in tissues, which binds to VEGF receptor 2 (VEGFR-2) on the surface of endothelial cells, triggering differentiation. Signaling through VEGFR-2 promotes the differentiation of tip cells and inhibits the formation of tip cells in neighboring cells through Notch signaling. VEGFR-2 is downregulated by Notch while VEGFR1 is overexpressed, sequestering VEGF and preventing over-vascularization [63]. Neighboring cells form the body of the sprouting vessel becoming stalk cells (Figure 3). Pharmacotherapy targets VEGF inhibits these processes and the eventual formation of leaking blood vessels. The phrase “anti-VEGF” encompasses a variety of distinct compounds, including aptamers (i.e., pegaptanib), antibodies to VEGF (i.e., bevacizumab), antibody fragments against VEGF (i.e., ranibizumab), and fusion proteins (i.e., aflibercept) [64] (Figure 4). A summary of findings from key clinical trials exploring these compounds for the treatment of DME are detailed in Table 1.

Pegaptanib sodium (Macugen^®^, Eyetech Pharmaceuticals, Melville, New York, NY, USA) is an anti-VEGF aptamer that selectively inhibits the action of VEGF165, the most physiologically relevant isotope of VEGF-A [71]. Pegaptanib was one of the first anti-VEGF inhibitors to undergo clinical trials in the treatment of age-related macular degeneration and DME. Pegaptanib’s effectiveness and safety in the treatment of DME were examined in a randomized, double-masked, multicenter, sham-controlled phase II study [72]. Three different dosages of pegaptanib were evaluated in comparison to sham: 0.3 mg, 1 mg, and 3 mg. All dosages of pegaptanib were well tolerated and showed improvements compared to sham. At 36 weeks, the median VA was significantly better with 0.3 mg (20/50) than with sham (20/63) (*p* = 0.04) with a notable reduction in central retinal thickness from 68 micron with 0.3 mg, compared to a four-micron rise with sham (*p* = 0.02). Additional studies have also shown similar results, with improvements in visual or anatomical outcomes [73,74,75,76]. However, due to the drug’s low clinical efficacy in comparison to other anti-VEGF medications, it is utilized considerably less often in clinical practice [77].

**Figure 3 cells-11-01950-f003:**
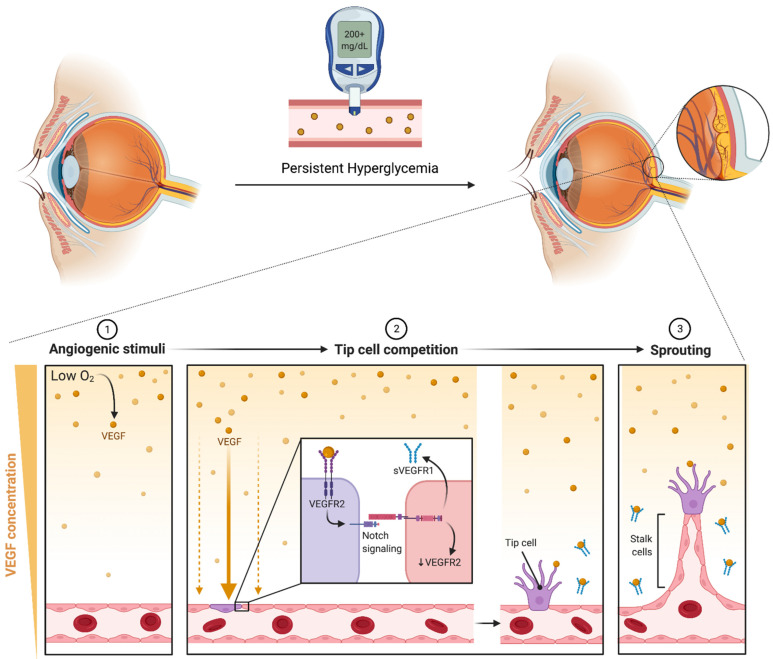
Persistent hyperglycemia causes retinal vascular structural changes that lead to hypoperfusion and decreased oxygen content. Low oxygen levels induce VEGF production in tissues, which binds to VEGF receptor-2 (VEGFR2) on the surface of endothelial cells triggering differentiation into tip cells. Inhibition of neighboring tip cell formation is carried out through notch signaling, which downregulates VEGFR2 and induces expression of soluble VEGF receptor-1 (sVEGFR1). sVEGFR1 serves to downregulate angiogenic VEGF-VEGFR2 signaling, thereby preventing excessive vascularization. Neighboring cells form the body of the sprouting vessel becoming stalk cells. It has been postulated that sVEGFR-1 functions as a guide molecule in vascular sprouting through inactivating VEGF on both sides of the sprout, creating a VEGF-rich path ensuring that the stalk grows in the correct direction [78,79] (created with BioRender).

Bevacizumab (Avastin^®^, Genentech, San Francisco, CA, USA) is licensed by the FDA for systemic therapy of metastatic colon cancer but not for use in the eye. It is used off-label in disorders such as age-related macular degeneration, diabetic retinopathy, and DME. Numerous studies have been conducted to determine the efficacy of bevacizumab in the treatment of DME. The intravitreal Bevacizumab Or Laser Therapy in the Management of Diabetic Macular Oedema (BOLT) research study was a two-year, prospective, single-center, randomized trial enrolling 80 patients with center-involving DME who had undergone at least one previous macular laser therapy [80]. The purpose of this study was to evaluate the effectiveness of repeated intravitreal bevacizumab therapy to that of four monthly modified macular laser treatments. At 12 months, the mean change in ETDRS VA in the laser group was −0.5 letters, but the bevacizumab group gained an average of 8 letters (*p* = 0.0002). Similar findings were obtained during the study’s 24-month follow-up [81]. Bevacizumab’s safety profile has been of some concern [82]. Intravenous bevacizumab has been linked with a number of systemic adverse events when used to treat some malignancies, including hypertension, proteinuria, and cardiovascular and gastrointestinal problems [83]. Ocular adverse events of bevacizumab include increased risks of endophthalmitis and ocular hypertension. In addition, bevacizumab has been associated with ocular inflammation [84].

Ranibizumab (Lucentis^TM^, Genentech, San Francisco, CA, USA) is a fragment of an antibody that also binds to and inhibits the action of VEGF and is approved for use in the US and the rest of the world for DME as well as other retinal diseases, including neovascular macular degeneration. In contrast to pegaptanib, ranibizumab binds to and suppresses all VEGF-A isoforms [85]. Several clinical trials have evaluated the efficacy of ranibizumab in the management of centers involving DME. 

**Figure 4 cells-11-01950-f004:**
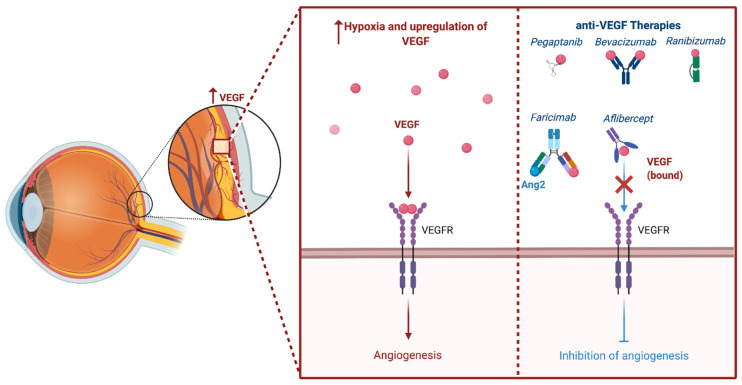
The phrase “anti-VEGF” encompasses a variety of distinct compounds that inhibit angiogenesis. The four most studied structural compounds include aptamers (i.e., pegaptanib), antibodies to VEGF (i.e., bevacizumab), antibody fragments against VEGF (i.e., ranibizumab), fusion proteins (i.e., aflibercept), and heteroantibodies (i.e., Faricimab) (created with BioRender).

The RESOLVE study was a one-year, phase 2, placebo-controlled, randomized, multicenter study evaluating the safety and effectiveness of ranibizumab in DME [65]. A total of 151 participants were randomly assigned to receive ranibizumab monotherapy (0.3 mg or 0.5 mg doses) or sham treatment. After three months, patients with continued disease activity were given rescue laser photocoagulation therapy. Patients had three consecutive monthly injections as initial therapy and were then followed monthly with an as-needed regimen from month three to one year. After one year it was discovered that the ranibizumab group achieved an average increase from baseline of 10.3 ± 9.1 letters compared to a 1.4 ± 14.2 letter drop in the sham group (*p* < 0.001). In addition, the mean central thickness reduction was 194.2 μm with ranibizumab (pooled data) vs. 48.4 μm with sham (*p* < 0.0001). There were no differences in the incidence of adverse events between ranibizumab and sham groups. To see whether 0.5 mg ranibizumab alone or active laser ±0.5 mg ranibizumab would be more effective, 345 patients took part in the multicenter phase 3 RESTORE study [66]. They found that, at 12 months, the mean average change in baseline VA letter score in the ranibizumab monotherapy group was +6.1, compared to +0.8 in the laser monotherapy group (*p* < 0.0001). Central macular thickness was also significantly reduced from baseline in the ranibizumab group (−118.7 μm) compared to the laser alone group (−61.3 μm) (*p* < 0.001). There were no observed significant changes in anatomical or visual function between ranibizumab alone group and ranibizumab with laser. RIDE and RISE are two similarly constructed, parallel, double-blind, three-year clinical trials that were 24 months long and placebo-controlled [67]. Seven hundred and fifty-nine patients were randomly assigned to undergo intensive monthly therapy with 0.3 mg ranibizumab, 0.5 mg ranibizumab, or placebo injection, with rescue laser available from month three. Patients who received ranibizumab were more likely to be able to read at least 15 ETDRS letters compared with sham group (*p* < 0.001). Additionally, the 0.3 mg ranibizumab group had an average letter increase of 12.5 ± 14.1 after 24 months, while the sham group had a 2.6 ± 13.9 increase (*p* < 0.001). Visual gains reported after 24 months of ranibizumab therapy were sustained with ongoing treatment for 36 months. Despite the fact that the sham cohort was switched over to ranibizumab, the gains in vision seen at the 36-month timepoint were significantly less than those observed in the ranibizumab-treated groups (*p* = 0.01) [86]. The RELIGHT trial assessed the potential advantages of adapting ranibizumab therapy to the individual requirements of DME patients [87]. Patients were administered a loading dose of three intravitreal ranibizumab injections. Patients received monthly intravitreal ranibizumab injections and bimonthly injections according to their individual VA and macular thickness. The findings of this trial indicated that functional outcomes gained during the first six months of therapy were sustained throughout the bimonthly individualized treatment. In terms of adverse events, ranibizumab has been associated with conjunctival hemorrhage, endophthalmitis, and ocular hypertension. A study by Bressler et al. [88] found that repeated intravitreal ranibizumab injections may raise the likelihood of prolonged intraocular pressure (IOP) increases or the requirement for ocular hypotensive therapy in eyes with center-involved diabetic macular edema and no prior open-angle glaucoma. However, a more recent review by Levin et al. [89] found that retinal nerve fiber layers (RNFL) were not diminished in eyes treated with ranibizumab in multiple studies, suggesting that regions of RNFL thinning in treated patients were more likely caused by the macular lesion than by the injections.

Aflibercept (Eylea; Bayer HealthCare, Whippany, NJ, USA) is another anti-VEGF drug that was authorized for intravitreal treatment in DME. Published data from two randomized Phase III studies (VISTA and VIVID) demonstrated that intravitreal aflibercept at 2 mg/0.05 mL was superior to macular laser in functional and anatomical outcomes, with equivalent effectiveness when administered monthly or biweekly. The mean VA gains from baseline to week 52 were 12.5 ± 9.5 and 10.7 ± 8.2 letters in VISTA and 10.5 ± 9.5 and 10.7 ± 9.3 letters in VIVID, respectively (*p* < 0.0001). In both the monthly and biweekly groups, the 100-week visual and anatomical superiority of aflibercept over laser control was maintained [68,90]. The Diabetic Retinopathy Clinical Research Network (DRCR.net) announced the Protocol T trial results, comparing anti-VEGF effectiveness and safety for DME [91]. Each patient was given either 2 mg aflibercept, 1.25 mg bevacizumab or 0.3 mg ranibizumab. Every 4 weeks, patients were re-injected if their condition deteriorated. The improvement in VA was significantly greater in the aflibercept group compared to bevacizumab and ranibizumab. Letter scores improved by 13.3, 9.7, and 11.2 for aflibercept, bevacizumab, and ranibizumab at 1 year, respectively (aflibercept vs. bevacizumab, *p* < 0.001; aflibercept vs. ranibizumab, *p* = 0.03). In addition, aflibercept was shown to lead to the greatest improvements in central subfield thickness (aflibercept vs. bevacizumab, *p* < 0.001; aflibercept vs. ranibizumab, *p* = 0.04). All therapies had nearly identical ocular and systemic safety profiles. Aflibercept was also shown to be more successful in improving eyesight in eyes with vision of fewer than 69 letters. The mean improvement with aflibercept was 18.9, bevacizumab was 11.8 and ranibizumab was 14.2. This study supports the effectiveness and safety of ranibizumab, bevacizumab, and aflibercept in DME patients, and recommends aflibercept for individuals with initial poorer eyesight [91]. Findings from the two-year results from the Protocol T trial found that mean VA improved by 12.8 ± 12.4, 10.0 ± 11.8, and 12.3 ± 10.5 letters for aflibercept, bevacizumab, and ranibizumab groups, respectively [92]. Aflibercept exhibited improved 2-year VA outcomes (18.1 ± 13.8, 13.3 ± 13.4, and 16.1 ± 12.1 letters, respectively) compared to bevacizumab in eyes with a lower baseline (VA 20/50 to 20/320), but the advantages of aflibercept seen after 1 year over ranibizumab were diminished (*p* = 0.18). 

A new VEGF therapy for DME is brolucizumab (Beovu^®^, Novartis, Basel, Switzerland), which is a single-chain antibody fragment. A brolucizumab dose of 6 mg produces a molar dosage that is approximately 11 times that of aflibercept (2 mg) and 22 times that of ranibizumab (0.5 mg) [64]. A study examining the off-label use of brolucizumab for three patients with recalcitrant DME found that, at 16 weeks, all three demonstrated notable improvement in BCVA (20/40 to 20/25) and a reduction in fluid on SD-OCT [93]. Results 52 weeks into the prospective, randomized, phase III clinical study in DME, KITE, which aimed to confirm brolucizumab (6 mg) noninferiority to aflibercept (2 mg), indicated that brolucizumab may provide robust vision gains and superior anatomic outcomes with q12w treatment intervals in more than 50% of patients with DME (10.6 letters for brolucizumab vs. 9.4 letters for aflibercept, *p* < 0.001) [93,94,95]. In the KESTREL study, subjects were randomized 1:1:1 to brolucizumab (3 mg and 6 mg) or aflibercept (2 mg). At 52 weeks, brolucizumab 6 mg was found to be non-inferior to aflibercept in terms of mean change in BCVA from baseline (9.2 letters for brolucizumab vs. 10.5 for aflibercept, *p* < 0.001) [96]. Findings from a report from an independent Safety Review Committee of the American Society of Retina Specialists (ASRS), which examined investigator-reported instances of intraocular inflammation, endophthalmitis, and retinal artery occlusion, showed that the incidence of intraocular inflammation (4.6%) was comparable to HAWK and HARRIER (4.4%), but the percentage of patients with retinal vasculitis (3.3%) and retinal vascular blockage (2.1%) was higher than it was in those studies (1%) [97]. The findings of the 1-year MERLIN trial were revealed by Novartis, and it was discovered that intraocular inflammation was greater with brolucizumab (9.3%) than with aflibercept (4.5%) [98]. 

Another emerging anti-VEGF therapy is Susvimo (Genentech), which is a surgically placed, refillable, ranibizumab-releasing port. Susvimo lasts 6 months and eliminates the need for monthly intraocular injections. It is the first implantable drug delivery system to deliver monoclonal antibodies [99]. A clinical trial (PAVILION) is currently underway evaluating the effectiveness, safety, and pharmacokinetics of the port delivery system with Ranibizumab compared to intravitreal ranibizumab 0.5 mg every 4 weeks. The estimated completion date for this trial is 2023. 

Anti-VEGF medicines are now regarded as the first line of therapy in center involving DME. Concerns remain regarding the routine use of broclizumab due to drug-induced retinal vasculitis risk. Nevertheless, a number of significant trials have indicated that, at most, 45% of DME patients treated with anti-VEGF medications improve by three lines or more [100]. In addition, a recent meta-analysis examined whether the frequency of anti-VEGF growth factor injections may be associated with a higher mortality risk due to the systemic absorption of the drug [101]. Authors found that, at 24 months, an increasing number of injections was related to marginally greater mortality risk in patients with DME (IRR = 1.17, *p* = 0.03). The lack of response to anti-VEGF in many individuals and a marginally increase in the risk of mortality with an increasing number of injections provides an impetus to explore supplementary or combinatorial therapies to enhance VA in patients with persistent DME.

### 3.4. Anti-VEGF Biosimilars

A biosimilar is a biologic medicinal product that is essentially a carbon replica of an original product made by a separate manufacturer [102]. Biosimilars are approved replicas of original founder products that may be made after the patent on the original product expires. One issue is that bevacizumab and ranibizumab’s US patents expired in 2019 and 2020, respectively, with Aflibercept’s patent expiring in 2023. A biosimilar molecule should be identical to the originator biologic in terms of pharmacodynamics, effectiveness, and safety [103]. There is, however, a significant distinction between generics and biosimilars, partly due to the need of live cells during the production processes. Studies have examined the effectiveness and safety profiles of biosimilars developed for ranibizumab, bevacizumab, and aflibercept. 

The first biosimilar developed from ranibizumab is Razumab (Intas Pharmaceuticals, Ahmedabad, India), which was approved in India in 2015. The RE-ENACT and CESAR studies evaluated the effectiveness of Razumab in Indian patients with several retinal pathologies, including DME [104,105]. A pooled analysis from the RE-ENACT study revealed that mean (SE) logMAR best-corrected VA scores improved from 0.75 ± 0.01 (Snellen equivalent ≈ 20/112) at baseline to 0.49 ± 0.01(Snellen equivalent ≈ 20/62) at the end of 12 weeks (*p* < 0.0001). In addition, the mean central macular thickness improved from 418.47 ± 4.78 μm at baseline to 301.17 ± 2.82 μm at the end of 12 weeks (*p* < 0.0001). The CESAR trial observed improvements in the mean logMAR corrected distance VA in patients with DME (0.60 ± 0.41 (Snellen equivalent ≈ 20/80) at baseline to 0.32 ± 0.32 (Snellen equivalent ≈ 20/42) at 3 months) (*p* < 0.001). The mean central foveal thickness was also found to improve from 436.70 ± 174.33 to 275.04 ± 120.09 μm in DME patients at three months (*p* < 0.001). After concluding these studies, there was no evidence of increased systemic side effects, elevated intraocular pressures, or signs of ocular toxicity. 

Another biosimilar development from ranibizumab is Ranizurel (Reliance Life Sciences Pvt Ltd., Mumbai, India). The Drugs Controller General of India approved Ranizurel in 2021 for neovascular age-related macular degeneration [106]. The Ranizurel safety evaluation in real-world (RaSER) study assessed the safety and efficacy after Ranizurel in 11 eyes of patients with DME, among other pathologies. They found that best-corrected VA improved from a logMAR baseline of 0.50 ± 0.27 (Snellen equivalent ≈ 20/63) to 0.30 ± 0.29 (Snellen equivalent ≈ 20/40) (a minimum of 4 weeks) (*p* = 0.004) [107]. Central subfield thickness was found to decrease from 473.4 ± 162.3 to 339.5 ± 96.9 μm (*p* = 0.003). The authors reported no associated ocular or systemic adverse effects.

There are numerous other biosimilar being studied for bevacizumab and aflibercept. Bevacizumab biosimilars are mostly used in cancer, but their off-label use for ophthalmic pathologies is increasing [103]. There are at least eight biosimilars of aflibercept. A double-masked, multicenter, phase-three clinical trial examining the effectiveness of MYL-1701P (Mylan N.V., Canonsburg, PA, USA), a biosimilar to aflibercept, for central DME was completed in 2021. Results from this trial have yet to be released. The expanded use of biosimilars for DME carries the benefit of lower treatment costs, increasing available options for physicians, and wider coverage. In developing nations where access to generic drugs can be limited, biosimilar pharmaceuticals have a unique opportunity to fill a need in medication access.

### 3.5. Intravitreal Corticosteroids

Leukocyte-induced inflammation has been shown to play a significant role in the development of diabetic induced vision loss in animal models [108]. There is evidence to suggest that leukocytosis may be epiphenomenon of the diabetic retinal environment, as opposed to an essential phase in the formation of human DR [109]. Steroids are nonspecific anti-inflammatory agents that play a key role in modulating inflammation through several mechanisms. In the retina, corticosteroids may act through stabilization of the blood–retina barrier, reduction in capillary permeability, and enhancing endothelial tight junction activity [110,111]. Currently available corticosteroids for the treatment of DME include intravitreal triamcinolone acetonide, dexamethasone intravitreal implant, and fluocinolone acetonide intravitreal inserts. 

Intravitreal triamcinolone acetonide (Kenalog) is currently an off-label treatment that is not approved for ocular use. Another formulation is approved for intraocular use, Triesence. The first reported use of intravitreal triamcinolone for DME was by Jonas and Sofker [112], where they intravitreally injected 20 mg of triamcinolone acetonide in a single patient with persistent DME following macular laser photocoagulation. They found that the patient’s vision improved from 0.10 to 0.40 to during the 5-month follow-up period, and the edema resolved. A study by Ciardella, et al. [113] found that 4 mg intravitreal injections of triamcinolone acetonide improved VA from 0.17 at baseline to 0.31 at 6 months. Multiple follow-up studies have shown similar short-term improvements in VA of patients with chronic DME that are unresponsive to conventional laser treatment [114,115,116,117,118,119,120,121]. Two DRCR.net randomized controlled trials investigated the use of intravitreal triamcinolone in the treatment of DME. In the first trial (Protocol B), intravitreal triamcinolone at doses of 1 mg or 4 mg was compared to focal/grid laser. Treatment with triamcinolone showed superiority in terms of VA at four months with a mean change in letters of +4 ± 12 in the 4 mg group compared to 0 ± 13 in the laser group (*p* < 0.001). However, at one-year follow-up, VA was comparable between laser and intravitreal triamcinolone treatments: +1 ± 16 for laser, 0 ± 15 for 1 mg, and 0 ± 16 for 4 mg. The two-year primary outcome was found to be +1 ± 17, −2 ± 18, and −3 ± 22 for the laser group, 1 mg triamcinolone group, and 4 mg triamcinolone group, respectively (*p* < 0.05) [122]. In the second trial (Protocol I), focal/grid laser alone was compared to 0.5 mg ranibizumab or 4 mg triamcinolone coupled with focal/grid laser. Treatment with triamcinolone plus laser after 24 weeks was found to confer better VA compared to laser alone. At 2 years, treatments between triamcinolone coupled with focal/grid laser (+2 ± 19) and focal/grid laser alone (+3 ± 15) showed equivalent outcomes (*p* = 0.35). However, in the triamcinolone group, they found that patients had significantly increased rates of elevated IOP (38%) and cataract surgery (15%) [123]. The findings from these studies have suggested that the beneficial effect of triamcinolone is short-lived, with no improvement compared to laser in the long term. In addition, cataract development and elevated eye IOP limit its long-term clinical use. Approaches to minimize adverse events through posterior sub-Tenon delivery are unlikely to be of substantial benefit [124]. A recent Phase I/II clinical trial found that multiple injections of triamcinolone into the suprachoroidal space were well-tolerated and achieved therapeutic drug levels in the retina while minimizing levels in the anterior parts of the globe [125]. 

The dexamethasone intravitreal implant (Ozurdex; Allergan Inc., Irvine, CA, USA) is an FDA-approved sustained-release bio-degradable implant formulation of corticosteroid for the treatment of DME. The implant is injected into the vitreous as an outpatient procedure. The implant provides a therapeutic effect for up to 4 months after injection, with peak activity at about 2 months. The MEAD study was a three-year Phase III randomized controlled trial of 1048 patients treated with repeated 0.7-mg dexamethasone implant, a 0.35-mg dexamethasone implant, or sham (control) injections [69]. Despite only 67% of patients in the dexamethasone group completing the study, the percentage of patients who gained ≥15 in letter score was 22.2%, 18.4%, and 12% in the 0.7 mg, 0.35 mg, and sham groups, respectively. Adverse effects were found to be similar to triamcinolone with approximately 60% of eyes in the 0.7-mg implant group having to undergo cataract surgery compared to 7% in the sham group over the 3-year study period. In addition, IOP was elevated in nearly 33% of eyes in the 0.7-mg implant group. 

Intravitreal inserts of fluocinolone acetonide (Iluvien; Alimera Sciences, Inc., Alpharetta, GA, USA) are cylindrical tubes of non-biodegradable polymer at varying doses. Inserts provide sustained delivery of fluocinolone acetonide for up to 3 years, which reduces the frequency of treatments and, thereby, may improve compliance and lowers the risk of injection-associated endophthalmitis [126]. The FAME was a controlled trial that investigated the use of Iluvien (Alimera Sciences Limited, Aldershot, UK) for patients with persistent DME [70]. Subjects were randomized to three groups: 0.2 μg per day (licensed implant dose), 0.5 μg per day, or sham injections. Patients were able to receive laser therapy 6 months after initial treatment if there was a failure to respond to steroids. At three years, the percentage of patients who gained ≥15 letters was 28.7%, 27.8%, and 18.9% in the 0.2 μg, 0.5 μg, and sham groups, respectively. Around 40% of eyes required rescue macular laser photocoagulation in the 0.2 μg. In addition, nearly all phakic patients in the treatment group developed cataracts, and nearly 40% of eyes in 0.2 μg/day arm required IOP to lower medications. A follow-up analysis aimed to correlate the duration of diagnosis of DME with treatment effect and found that patients who had DME for >3 years showed a greater response to the fluocinolone acetonide than patients who had DME for a shorter duration (34.0% vs. 22.3%, respectively) [127]. 

At present, intravitreal steroid implants are considered a second-line treatment after anti-VEGF. They may also be primarily used following certain indications, including chronic DME and in patients where anti-VEGF may be unsafe, such as during pregnancy. 

### 3.6. Combinatorial Therapy

There is growing interest in the use of combination therapy to treat DME. Despite the fact that anti-VEGF therapy leads to good structural and optical outcomes for a large number of patients, many need injections for an extended length of time to preserve these benefits. Combination therapy is primarily intended to lengthen the duration of the anti-VEGF action and minimize the number of injections, which may assist in reducing the treatment burden and improving patient safety. 

A number of studies have compared anti-VEGF + conventional macular laser treatment to anti-VEGF alone. While most studies show no statistically significant difference in functional and anatomical results between monotherapy and combination treatment groups, the latter was linked to fewer anti-VEGF injections [67,128,129]. Several studies have also explored outcomes from the combination of Subthreshold diode micropulse laser + anti-VEGF therapy [130,131,132]. Luttrull et al. (2012) explored the micropulse laser + anti-VEGF in patients with DME. The authors observed a significant reduction in central foveal thickness in 71% of the micropulse-alone group and 89.5% in the combination group, although there was no significant difference between treatment arms (*p* = 0.16). The best-corrected VA was stable overall and in both the micropulse-alone and combination treatment groups. Thinda et al. (2014) found overall similar results, with patients who underwent additional micropulse photocoagulation having a lower frequency of anti-VEGF injections, with no significant change in anatomical or functional results. These findings suggest that combination therapy with anti-VEGF injections and micropulse laser may primarily benefit patients primarily lowering the treatment burden. Navigational laser in conjunction with anti-VEGF may provide an extra advantage over conventional laser through the quicker clearance of macular edema. 

Intravitreal triamcinolone has been examined in combination with conventional and subthreshold micropulse lasers. Kang, et al. [133] evaluated the clinical outcomes of macular laser after the intravitreal injection of 4 mg of triamcinolone compared to triamcinolone monotherapy for diffuse DME. At 6 months, the mean logMAR VA 0.71 ± 0.41 (Snellen equivalent ≈ 20/103) in the combined group and 1.06 ± 0.45 (Snellen equivalent ≈ 20/230) in the control group (*p* < 0.001). Lam, et al. [134] explored the outcomes of intravitreal triamcinolone plus sequential grid laser versus triamcinolone or laser alone in patients with DME. Authors observed that triamcinolone combined with laser produced the greatest reduction in central macular thickness vs. laser alone. The best central foveal thicknesses were achieved at 4 weeks in the intravitreal triamcinolone and combined groups and were 267 ± 75 μm and 256 ± 73 μm, respectively. However, no significant no difference in best-corrected VA was observed between the groups at any timepoint. Another study that examined outcomes after 24 months of intravitreal triamcinolone plus laser vs. laser treatment only in eyes with DME [135] found that there was no observed difference in the mean central macular thickness or mean logMAR VA between treatment groups. Additionally, combined treatment was associated with higher rates of cataracts and elevated intraocular pressure. This brings up the primary disadvantage of triamcinolone as part of a combined treatment approach for DME, which is its short duration of effect and the need for frequent injections, increasing the risk of cataracts and glaucoma. However, the increasing availability of corticosteroid implants has enabled novel combinatorial techniques in treating DME. 

The PLACID trial compared combination therapy of dexamethasone implant and modified ETDRS macular laser to the macular laser alone [136]. Macular laser treatment was initiated one month after dexamethasone implant treatment with no re-treatment with another dexamethasone implant until six months. They found that the percentage of patients who gained 10 letters or more in best-corrected VA at month 12 was not different between treatment arms, while the percentage of patients who gained 10 or more letters was significantly greater in the combination group at 1 month (*p* < 0.001) and 9 months (*p* = 0.007). It is possible that the study protocol limited the clinical effectiveness of the dexamethasone implant and that a more favorable outcome could have been achieved in the combination treatment arm if early re-treatment had been permitted at a 4-month timepoint, when the dexamethasone implant effect was expected to wear off.

Another study investigated the efficacy of combined dexamethasone intravitreal implant (0.7 mg) combined with bevacizumab (1.25 mg) vs. bevacizumab monotherapy [137]. If the OCT was larger than 250 µm and visual acuity was less than 6/9, eyes in the bevacizumab-only group received monthly therapy. Similarly, the combination therapy group received monthly bevacizumab as needed, but additionally received dexamethasone implants at months 1, 5, and 9. At one year, both groups improved similarly in terms of visual acuity; however, the combination therapy group considerably improved in terms of retinal thickness reduction (−45 µm vs. −30 µm, *p* = 0.03). No significant difference in treatment load was seen between the two groups, since the combination therapy group received three fewer bevacizumab injections than the bevacizumab group over a one-year period, in addition to two extra dexamethasone implants.

PALIDIN was a phase 4, nonrandomized, open-label observational study that included patients with DME who received a fluocinolone acetonide intravitreal implant at baseline and then were observed for up to 36 months [138]. Combination treatment was used including anti-VEGF therapy. At 36 months after fluocinolone implant, the study eyes showed a mean CST change of −60.69 μm (*p* < 0.0001) and a mean BCVA change of +3.61 letters. Median treatment frequency decreased from 3.4 treatments/year in the 36 months before the implant to 1 treatment/year in the 36 months after the implant. 

### 3.7. Pars Plana Vitrectomy

Since the 1980s, the vitreous has been thought to play a major role in DME pathogenesis [139]. Variations in vitreous humor molecular profiles in patients with DME may provide additional insights into the mechanism of pathogenesis. It has been suggested that chronically elevated blood sugar levels lead to the glycation of collagen fibrils in the vitreous humor, causing abnormal crosslinks to develop [140]. In addition, blood–retinal barrier breakdown allows for cytokine accumulation in the vitreoretinal interface, leading to collagen matrix destabilization and prompting tangential traction and macular edema [141]. The effectiveness of particular intravitreal treatments can decrease with pathology at vitreoretinal interface [142]. The induction of posterior vitreous detachment via pars plana vitrectomy can be used to treat DME in setting concurrent vitreomacular traction. Vitrectomy is thought to eliminate traction through mechanical and non-mechanical mechanisms. Mechanically, vitrectomy eliminates traction by removing epiretinal membranes, internal limiting membranes, and posterior hyaloid [140]. Non-mechanically, vitrectomy has been shown to increase oxygen tension in the vitreous cavity and may downregulate VEGF [143]. 

Patients with DME with vitreomacular traction have been shown to benefit from vitrectomy. A prospective cohort study conducted by DRCR.net of 87 eyes with DME and concurrent vitreomacular traction found that pars plana vitrectomy reduced retinal thickening by 50% in 68% of eyes [144]. VA improvements were observed in in 38% of eyes, with a notable gain of ≥10 letters. In the absence of vitreomacular traction, the role of vitrectomy is not as clear. Thomas et al. [145] found that patients with DME who had previously received laser did not benefit from vitrectomy in terms of increased VA or decrease in macular thickness. In addition, a meta-analysis conducted by Simunovic et al. (2014) found that at 12 months vitrectomy has no structural benefit and inferior functional outcomes compared to laser, with minimal evidence recommending vitrectomy as a treatment for diabetic macular edema in the absence of an epiretinal membrane or vitreomacular traction [146]. It has been suggested that, since vitrectomy was used as a last resort in many of these studies after treatment failure with lasers or intravitreal injections, it is unsurprising that visual results were poor. A more recent study reported the outcomes in patients undergoing vitrectomy with internal limiting membrane peeling for the management of treatment-naïve DME [147]. The authors found that, at 6 months follow-up, VA improved from 0.74 logMAR (Snellen equivalent ≈ 20/110) at baseline to 0.46 logMAR (Snellen equivalent ≈ 20/58) (*p* = 0.045). In addition, they observed a significant decrease in macular thickness with surgery, with central macular thickness decreasing from 456 µm to 316.8 µm (*p* < 0.001). These findings indicate that patients with a less complicated, shorter disease course may benefit from primary vitrectomy treatment.

At present, vitrectomy is mainly reserved for DME cases where there is vitreomacular traction and a lack of an anti-VEGF treatment response [144]. A critical practical consideration when managing DME patients who have previously undergone vitrectomy surgery is that enhanced drug clearance from the vitreous cavity may result in a slower rate of anatomic improvement following anti-VEGF treatment, with more frequent injections required, compared to eyes that have not undergone vitrectomy [148]. However, the half-life of steroid implants such as dexamethasone and fluocinolone acetonide is not shortened after vitrectomy. 

## 4. Novel Therapeutics

While many patients’ DME is well-controlled with the current anti-VEGF therapy, there is evidence that VEGF may not be the main mechanism promoting vascular permeability in DME, so there is considerable interest in developing therapies that target different mechanisms in order to increase efficacy [149]. Recent clinical trials have looked at how antibodies targeted against cytokine/chemokines, adhesion molecules, and multiple growth factors could offer therapeutic benefits in patients with DME, both as a standalone treatment or used in combination with current primary treatment [100]. The current novel therapeutics being investigated in the treatment of DME are summarized in Figure 5.

### 4.1. Cytokine Inhibitors

In patients with DME, levels of Angiopoietin-2 (Ang2) have been found to be significantly augmented in the vitreous [150]. The Angiopoietin–Tie (Ang/Tie) signaling pathway is a vascular specific receptor tyrosine kinase pathway involved in vascular development. Members of the Ang/Tie pathway play important roles in endothelial physiology and have been associated with retinal microvascular diseases. It has been reported that the co-expression of Ang-2 and VEGF-A accelerates neovascularization in developing retina and ischemic retina models [151].

In January of 2022, Faricimab-svoa (Vabysmo^TM^; Genentech, San Francisco, CA, USA) was approved by the FDA to treat neovascular age-related macular degeneration and diabetic macular edema. Faricimab is an antibody that targets the Ang2 and VEGF-A receptors. Faricimab is a heteroantibody comprised of two different heavy and light chains with bispecific properties: one heavy and light chain binds to Ang2 and the other to VEGF-A (Figure 4) [152]. By targeting both Ang2 and VEGF, faricimab is able to reduce the number of vascular lesions, permeability, retinal edema, and neuron loss, thereby increasing the effect from the dual inhibition versus single inhibition of either factor [152]. The dual inhibition provides an advantage for faricimab compared to monotherapy, and therefore likely has a higher chance of success in real-world uses. The BOULEVARD trial was a phase II clinical trial that compared the safety and efficacy of faricimab with ranibizumab in patients with DME for a study period of 36-weeks across 59 sites in the United States [153]. Patients in this study were treated with either intravitreal 6.0 mg faricimab, 1.5 mg faricimab, or 0.3 mg ranibizumab if previously treatment-naïve, or intravitreal 6.0 mg faricimab or 0.3 mg ranibizumab if previously treated with anti-VEGF therapy. Eyes were dosed once a month for 20 weeks, followed by an observation period up to week 36, to assess durability. This study showed that faricimab demonstrated statistically superior VA gains of 3.6-letter mean vision gain (compared to ranibizumab) at week 24 in treatment naïve patients (*p* = 0.03; 80% CI, 1.5–5.6 letters). The study also showed central subfield thickness (CST) reduction, increases in Diabetic Retinopathy Severity Scale (DRSS) score, and extended durability outcomes that support the primary outcome measure of mean VA change from baseline. These findings support that there is a potential clinical benefit when simultaneously inhibiting both Ang2 and VEGF-A in patients with DME [153].

After success in the phase II clinical trials, two phase III clinical trials, YOSEMITE and RHINE were conducted. Both studies are identical, randomized, multicenter, double-masked, global phase III studies, that seek to evaluate the safety and efficacy of faricimab compared to aflibercept in a total of 1891 people with DME [154]. Each study was conducted with three treatment arms: faricimab 6.0 mg administered at personalized dosing intervals of up to four months; faricimab 6.0 mg administered at fixed two-month intervals; aflibercept 2.0 mg administered at fixed two-month intervals (Eter et al., 2021 [154]; Sharma et al., 2021 [152]). Since the trials were conducted using a novel personalized treatment intervals (PTI) regimen, meaning the trial allows for a variable injection frequency from 1 month to 4 months depending on the activity of the patient’s disease, each of the arms of both trials maintained masking through the use of sham injections that were administered to participants at visits when treatment injections were not scheduled [154]. The one-year results for faricimab showed, on average, VA gains of 11.6 (97.52% CI 10.3 to 12.9) and 10.7 (97.52% CI 9.4 to 12.0) letters in the PTI and every other month arms, respectively, and an average gain of 10.9 letters in the aflibercept arm in the YOSEMITE study (Wykoff et al., 2022 [154]). In the RHINE study, results were similar, showing mean VA gains of 10.8 (97.52% CI 9.6 to 12.0) and 11.8 (97.52% CI 10.6 to 13.0) letters in the PTI and every other month arms, respectively, for faricimab, and an average gain of 10.3 (97.52% CI 9.1 to 11.4) letters in the aflibercept arm [155]. The results showed that more than 70% of patients on the faricimab PTI regimen were able to go ≥3 months between each treatment interval and treatment was well-tolerated in both clinical trials [155]. The study is currently following participants and will release year-2 data when available.

It should be noted that BOULEVARD, YOSEMITE, and RHINE trials compared faricimab to anti-VEGF therapies (i.e., ranibizumab and aflibercept) and did not compare to Ang-2 inhibition as monotherapy. Notably, higher molar doses of anti-VEGF were given in the faricimab group across trials. There were more ocular inflammation events with faricimab in the two-month and PTI arms compared to the aflibercept two-month arm. All intraocular inflammation events were cured by week 56, with the exception of two cases. Considering these findings, and the fact that YOSEMITE and RHINE trials showed that faricimab was non-inferior to aflibercept at the one-year endpoint, the clinical benefits of additional Ang-2 inhibition compared to anti-VEGF monotherapy is not clear at this point.

### 4.2. Adhesion Molecule Inhibitors

Integrins are cell-surface receptors that control attachments between cells, dictating communication in the extracellular matrix between the cells in the vicinity. In the retina, integrins are located on endothelial cells that make up the microvasculature, where they interact with various growth factors and help to regulate their functions [156].

Risuteganib (Luminate^®^, Allegro Ophthalmics, San Juan Capistrano, CA, USA) is an integrin antagonist that is currently being reviewed in clinical trials as a treatment candidate for DME and dry AMD. Risuteganib antagonizes four different integrin heterodimers, including αVβ3, αVβ5, α5β1, and αMβ2, which have been shown to be involved in the pathologic process of retinal angiogenesis (Shaw et al., 2020 [156]). While this drug is being targeted to treat both AMD and DME, the DME arm of the Phase II trial showed that risuteganib injections administered monthly for three months following initial bevacizumab injection was noninferior to monthly bevacizumab and was shown to be effective through 12 weeks after the final dose (48% of patients gained ≥8 letters in the treatment arm, compared to 7% in the sham [156]. It is important to note that the data thus far are short-term, and long-term effects have yet to be proven stable in patients with DME, although, if found to be safe and effective in the long-term it could provide a useful drug for the treatment of DME.

Vascular adhesion protein-1 is an endothelial adhesion molecule, which is expressed on the surface of vascular endothelial cells. It has been shown to be involved in leukocyte trafficking during inflammatory processes. ASP8232 (Astellas; Northbrook, IL, USA) is an orally administered VAP-1 inhibitor used in DME patients in the phase II VIDI trial designed to evaluate its safety and efficacy in reducing excess retinal thickness when given alone or in combination with ranibizumab in patients with center involving DME (CI-DME) (clinicaltrials.gov-VIDI trial). The mean increase in VA score from baseline in the ASP8232 group was 3.1 ± 7.3, in the ASP8232/ranibizumab group was 5.2 ± 7.1, and in the ranibizumab group this was 8.2 ± 9.5 (*p* = 0.015). This trial showed that ASP8232 alone was not efficacious in reducing CST in patients with DME compared to ranibizumab (*p* = 0.108). While the drug was shown to be able to almost completely inhibit VAP-1 activity, both monotherapy and combination therapy with anti-VEGF drugs did not provide any added benefits in eyes with DME. Since this drug was systemically administered by an oral route, future research could target local delivery to the eye to assess if efficacy could be reached in this way.

Another integrin inhibitor is THR-687 (Oxurion NV, Leuven, Belgium), which is a highly selective, arginine–glycine–aspartic acid (RGD) antagonist that is being developed as a treatment for DME patients. THR-687 has proven promising in animal models, where it was recently shown to reduce inflammation, vascular permeability, and reactive gliosis in diabetic rats [157]. A phase 1, open-label, multicenter dose-escalation study examined changes from baseline BCVA at 3-months follow-up [158]. In this trial, 3 patients were administered with 0.4 mg THR-687, 3 patients were administered 1.0 mg THR-687, and 6 patients were administered 2.5 mg THR-687. No dose-limiting toxicities or serious adverse events were reported at any dosage level. At day 7, there was a mean rise of 7.2 letters in BCVA reported (95% CI 4.1–10.3 letters). At 1 month, there was mean gain of 9.2 letters (95% CI 5.1–13.2 letters). The improvement in BCVA was maintained until the end of the trial at month 3, with a mean increase of 8.3 letters (95% CI 4.0–12.0 letters). Three individuals needed rescue therapy, including one patient treated with the 1.0 mg dosage of THR-687 and two patients treated with the 2.5 mg dose. The mean CST at baseline was 541.8 µm, with the highest-dose group having a lower CST (499 µm) than the two lower-dose groups (557 µm and 612 µm). The efficacy of THR-687 in comparison to the standard of care therapies has yet to be determined. A two-part phase two clinical trial is underway by Oxurion (“INTEGRAL”), which will evaluate the efficacy and safety of THR-687 vs. aflibercept in Part B of the trial in treatment-naïve and treatment-experienced DME patients (https://www.clinicaltrials.gov/ct2/show/NCT05063734; accessed on 6 June 2022). 

### 4.3. Kallikrein–Kinin (KK) System Inhibitors

Blood plasma prekallikrein (PPK) is an abundant serine protease zymogen that is transformed by factor XIIa to its catalytically active form, plasma kallikrein (PK) [159], where it contributes to inflammatory responses. PK serves to cleave kininogen, which generates bradykinin. Bradykinin (BK) is a nonapeptide that stimulates the BK receptors, which are expressed in high abundance in vascular, glial, and neuronal cell types [160]. BK receptor activation induces vasodilation and increases vascular permeability [161]. Proteomic analyses of vitreous samples from patients with DME have shown that PPK and PK are elevated 11.0-fold compared to control vitreous from patients with macular holes (*p* < 0.0001) [162]. The systemic, continuous administration of a PK inhibitor in diabetic mice was shown to diminish retinal vascular permeability and retinal blood flow abnormalities [163]. Additional preclinical evidence has demonstrated that plasma kallikrein may contribute to VEGF-mediated causes of retinal edema [164].

KVD001 (KalVista Pharmaceuticals, Boston, MA, USA) is a plasma kallikrein inhibitor that is currently being developed as an intravitreal treatment. The safety, pharmacokinetics, and efficacy of KVD001 were assessed in a phase 1, open-label, single-ascending-dose clinical trial [165]. This study included diabetic patients with central involved DME and a BCVA of 18 to 68 letters. On days 1, 7, 14, 28, and 56 following a single administration of 1 µg (*n* = 3), 3 µg (*n*= 3), or 10 µg (*n* = 8), BCVA, intraocular pressure, dilated fundus examination, spectral-domain OCT, and a review of the adverse events were performed. A single dosage of KVD001 resulted in mean VA improvements of 0.7, 1.0, 1.9, 2.8, and 4.1 letters compared to baseline at days 7, 14, 28, 56, and 84, respectively, in 12 patients who completed the 12-week trial. Notably, 71% of patients reported at least 1 adverse event, with 3 patients in the low-dose group (1 µg) and 8 patients in the high-dose group (10 µg). Three patients in the mid-dose group (3 µg) showed moderate ocular irritation. 

A phase 2, sham-controlled, double-masked clinical trial examined KVD001 at two doses (6 µg and 3 µg) injected monthly over three months in 129 patients with center involving DME who were VEGF-experienced (https://clinicaltrials.gov/ct2/show/NCT03466099; accessed on 7 June 2022) [166]. The 6 µg dosage of KVD001 demonstrated a +2.6-letter BCVA improvement compared to placebo; however, the findings did not achieve statistical significance (*p* = 0.223). Compared to placebo, the lower dose of KVD001 resulted in a difference of +1.5 letters (*p* = 0.465). Despite these uncertain findings, approximately 32% of patients treated with the high-dose group showed progressive vision loss compared to 54% in the placebo group (*p* = 0.042), indicating that KVD001 may have a potential protective effect against the loss of vision. 

THR-149 (Oxurion NV, Leuven, Belgium) is a bicyclic peptide inhibitor of human plasma kallikrein. A phase 1, open-label, multicenter dose-escalation study with 3-month follow-up evaluated the safety and efficacy of a single intravitreal injection of THR-149 at three dose levels (5, 22, 130 µg) in 12 DME patients [167]. All subjects completed the study and no serious adverse events were recorded. There was a mean change from Baseline BCVA of +7.5 letters on Day 14, and +6.4 letters by the third month. The average CST change from baseline at month 3 was +30.0 µm. 

In early clinical trials, therapies that modulate the PK pathway for the treatment of DME yielded mixed outcomes. Further studies examining the efficacy of PK inhibitors, compared to and in combination with, the standard of care, are warranted. Oral PK inhibitors are also being developed for DME, which, if effective, could play a role in decreasing physician and patient burden [166]. Additional research will aid in determining whether oral PK inhibitors could play a role in DME. 

## 5. Conclusions

This anti-VEGF therapy remains the primary therapeutic approach for DME involving the center of vision. Macular focal/modified grid lasers continue to have a place during instances of clinically progressive non-center involving DME. While anti-VEGF therapy has been shown to lead to excellent structural and visual success, it has placed a strain on patients and health care providers, as frequent intravitreal injections are typically required for long-term vision maintenance. There is renewed interest in combining anti-VEGF therapy with other treatment modalities to aid in consolidating the effect of anti-VEGF therapy on the retina, and thus reducing the burden of repeat treatments, such as macular laser or sustained-release steroid implants. Combining laser and anti-VEGF therapy may help to alleviate some of the strain associated with anti-VEGF repeat treatments, but this combination has not been shown to lead to significant visual improvements. There is a dearth of data in the current literature examining the use of combined anti-VEGF and corticosteroid implants, and the duration of intervention. In addition, the risk of cataracts and glaucoma may limit its use. Patients who are pseudophakic, particularly those who require repeated anti-VEGF injections or are resistant to anti-VEGF medication, may be suitable candidates for combination therapy with corticosteroids. This has provided added pressure for novel therapeutic approaches. In recent years, research has shed light on a variety of growth factors and inflammatory markers that may play a role in DME pathophysiology. These compounds are being explored in tandem with anti-VEGF medicines such as monotherapy. Faricimab, an antibody that targets the Ang2 and VEGF-A receptors, may help extend the treatment duration to ≥3 months. Future clinical studies should be able to provide sufficient evidence to justify the use of these innovative medicines and treatment approaches.

## Figures and Tables

**Figure 1 cells-11-01950-f001:**
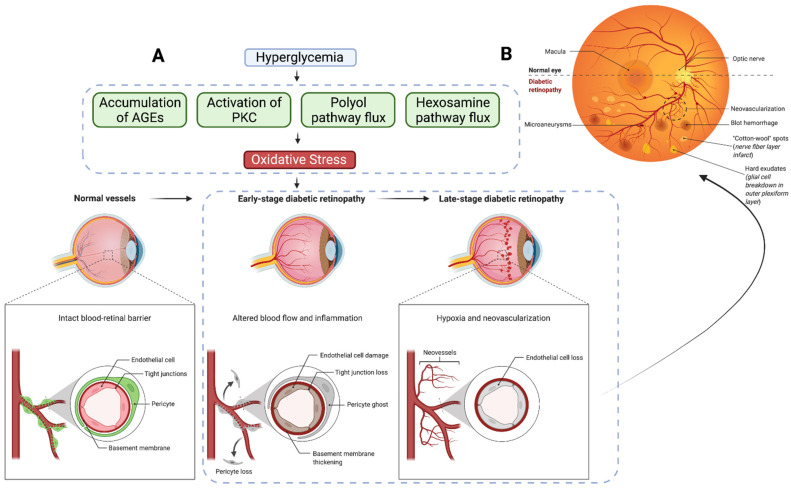
(**A**) Chronic hyperglycemia in diabetes leads to the upregulation of pathways, activation of enzymes, and accumulation advanced glycosylated end products. Each of these pathways can incite oxidative stress and set off the pathogenesis of diabetic retinopathy, which is characterized by loss of vascular endothelial cells, tight junctions, and pericytes, with basement membrane thickening, ultimately leading to hypoxia and neovascularization. (**B**) This can be observed in the retina as dot blot hemorrhages, cotton wool spots, and hard exudates in the outer plexiform layer (created with BioRender).

**Figure 2 cells-11-01950-f002:**
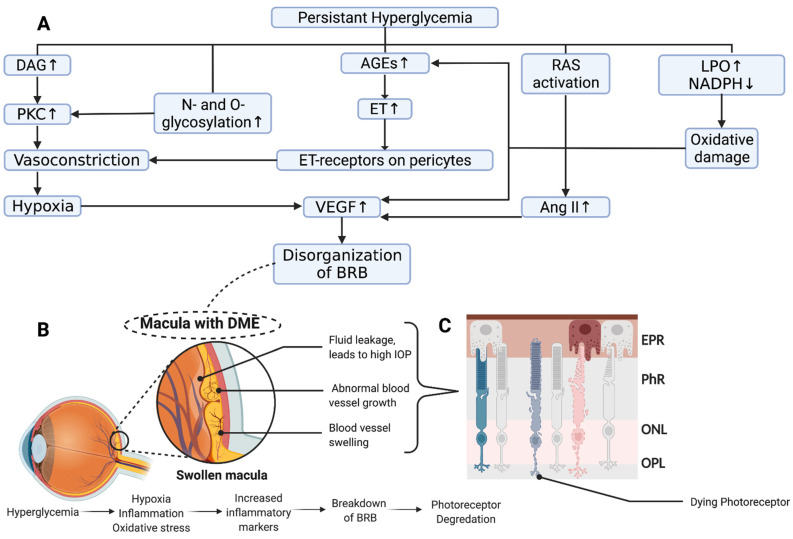
(**A**) Schematic overview of potential mechanisms leading to diabetic macular edema. Hyperglycemia-induced metabolic stress leads to a complex interaction leading to vascular damage to and compromise of the blood–retinal barrier (BRB) (Modified from [29]). (**B**) Disorganization of the BRB and hypoperfusion causes fluid extravasation, neovascularization, and subsequent edema. (**C**) Chronic swelling of the macula leads to damage to the neural retina and loss of photoreceptors (created with BioRender).

**Figure 5 cells-11-01950-f005:**
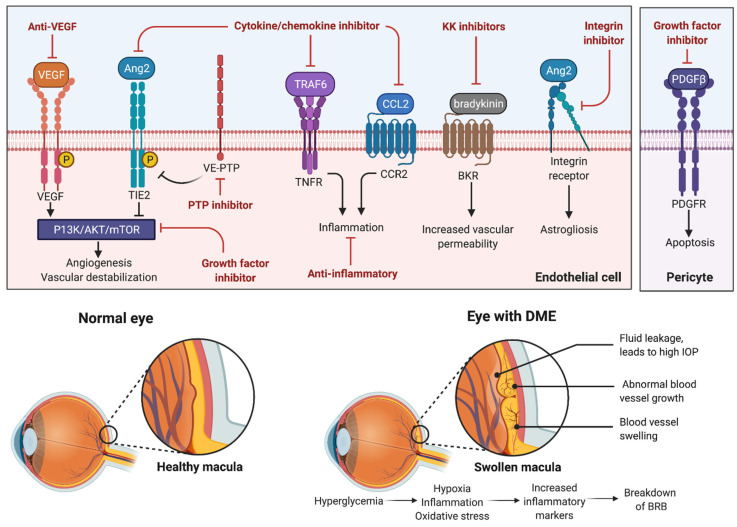
This figure provides an overview of innovative therapeutic approaches that target distinct signaling pathways implicated in the pathophysiology of DME.

**Table 1 cells-11-01950-t001:** Summary of key clinical trials on visual outcomes following intravitreal anti-vascular endothelial growth factor for diabetic macular edema.

					BCVA (ETDRS Letters)		CRT (µm)	
Study [Ref.]	Regimen	No. of Patients	Study Design	Group	Baseline	Change	Baseline	Change
RESOLVE [65]	Rani 0.3 mg	Control: 49	Phase II, double-masked, randomized, sham- controlled, multicenter	Rani	60.2 (9.9)	7.8 (7.7)	455.4 (114.2)	−194.2 (135.1)
	Rani 0.5 mg	Total: 151		Sham	61.1 (9.0)	−0.1 (9.8)	448.9 (102.8)	−48.4 (153.4)
RESTORE [66]	Rani 0.5 mg	Control: 111	Phase III, double-masked, randomized, laser- controlled, multicenter	Rani + sham	N.A.	6.1 (6.3)	N.A.	−118.7 (115.1)
	Rani 0.5 mg + laser	Total: 345		Rani + laser	N.A.	5.9 (7.9)	N.A.	−128.3 (114.3)
				Laser + sham	N.A.	0.8 (8.6)	N.A.	−61.3 (132.3)
RISE [67]	Rani 0.3 mg	Control: 127	Phase III, randomized, sham-controlled, multicenter	Rani 0.3 mg	54.7 (12.6)	12.5	474.5 (174.8)	−250.6
	Rani 0.5 mg	Total: 377		Rani 0.5 mg	56.9 (11.6)	11.9	463.8 (144.0)	−253.1
				Sham	57.2 (11.1)	2.6	467.3 (152.0)	−133.4
RIDE [67]	Rani 0.3 mg		Phase III, randomized, sham-controlled, multicenter	Rani 0.3 mg	57.5 (11.6)	10.9	482.6 (149.3)	−259.8
	Rani 0.5 mg			Rani 0.5 mg	56.9 (11.8)	12.0	463.8 (175.5)	−270.7
				Sham	57.3 (11.2)	2.3	447.4 (154.4)	−125.8
VISTA [68]	Laser	Laser: 154	Phase III, double-masked, randomized, active- controlled, multicenter	Aflib. 2q4	58.9 (10.8)	11.5	485 (157)	−191.4
	Aflib. 2q4	Aflib: 307		Aflib. 2q8	59.4 (10.9)	11.7	479 (154)	−191.1
	Aflib. 2q8			Laser	59.7 (10.9)	6.3	483 (153)	−83.9
VIVID [68]	Laser	Laser: 133	Phase III, double-masked, randomized, active- controlled, multicenter	Aflib. 2q4	60.8 (10.7)	11.8	502 (144)	−211.8
	Aflib. 2q4	Aflib: 271		Aflib. 2q8	58.8 (11.2)	10.6	518 (147)	−195.8
	Aflib. 2q8			Laser	60.8 (10.6)	5.5	540 (152)	−85.7
MEAD [69]	DEX 0.7 mg	Control: 350	Phase III, masked, randomized, sham-controlled, multicenter	DEX 0.35 mg	56.1 (9.9)	18.4% ^a^	463.0 (157.1)	−107.9 (135.8)
	DEX 0.35 mg	Total: 1048		DEX 0.7 mg	55.5 (9.7)	22.2% ^a^	466.8 (159.5)	−111.6 (134.1)
				Sham	56.9 (8.7)	12.0% ^a^	460.9 (132.6)	−41.9 (116.0)
FAME [70]	IFSR 0.2 µg/day	Control: 185	Double-masked, randomized, sham- controlled, parallel-group, multicenter	IFSR 0.2 µg	53.3 (12.7)	*p* = 0.019	460.8 (160.0)	*p* ≤ 0.003
	IFSR 0.5 µg/day	Total: 953		IFSR 0.5 µg	52.9 (12.2)	*p* = 0.015	460.8 (160.0)	*p* ≤ 0.003
				Sham	54.7 (11.3)	N.A.	460.8 (160.0)	N.A.

Note. Data are expressed as mean (standard deviation). BCVA, best-corrected visual acuity; EDTRS, Early Treatment of Diabetic Retinopathy Study; Rani, ranibizumab; Afib, aflibercept; 2q4, every 4 weeks; 2q8, every 8 weeks; IFSR, intravitreal fluocinolone sustained-release; DEX, dexamethasone implant. ^a^ Proportion of patients after three years with a ≥15-letter improvement in BCVA from baseline.

## Data Availability

Not applicable.

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
