# Peer review of "Current and Novel Therapeutic Approaches for Treatment of Diabetic Macular Edema"

_cells, 2022, doi:10.3390/cells11121950_

Round 1

Reviewer 1 Report

This review is complet and well organized.

I suggest:

1. add one or more table/s to summarize the studies discussed

2. faricimab has now been approved by the FDA. Please add this information in the section

3. I would delete the paragraph on squalamine 

4. minor English revision is needed

Author Response

Re: Manuscript ID cells-1759286 entitled " Current and Novel therapeutic Approaches for Treatment of Diabetic Macular Edema"

*All co-authors have reviewed and agree with the following responses and revisions to the manuscript.

Reviewer 1

Comment 1: This review is complete and well organized.

Response 1: Thank you very much for taking the time to review our work and for your positive comments.

Comment 2: Add one or more table/s to summarize the studies discussed.

Response 2: Thank you for this suggestion, we have added in Table 1, which summarizes findings from key (level 1) clinical trials.

Comment 3: Faricimab has now been approved by the FDA. Please add this information in the section.

Response 3: We have added in details of FDA approval (Lines 723-724).

Comment 4: I would delete the paragraph on squalamine.

Response 4: We have removed the paragraph on squalamine. In replacement, we have added a new section on KVD001 and THR-149 (kallikrein inhibitors), as these trials have recently been completed for DME.

Comment 5: Minor English revision is needed.

Response 5: Thank you for your suggestion. We have made grammatical improvements throughout.

Reviewer 2 Report

The review gives a nice overview of the current treatments for DME and includes beautiful illustrations. It is well written and it discusses the most important findings and new developments.

Below, some suggestions to improve the manuscript, in chronological order.

Abstract and Line298, Line 699: the abbreviation for vascular endothelial growth factor is spelled wrong

Abstract: angiogenesis and neovascularization are mentioned in the same sentence, but have the same meaning.

Abstract: need, perspective or novelty of this review should be mentioned to increase its value. The part that is at the end of the Introduction now.

Introduction: Type 2 diabetes mellitus was previously called non-insulin dependent diabetes mellitus (NIDDM) and late onset diabetes mellitus. These names are no longer used because they are inaccurate. Insulin is often used in the management of type 2 diabetes.

Introduction: blindness caused by DM not only occurs in the US, also in other parts of the world. It will be appreciated if one looks a little further than just the US. Please add or replace reference.

Introduction: Ref17 does not cover results of PKC-induced hypoxia and blood-retinal barrier (BRB) impairment. Please replace.

Figure 1B: if possible, increase font size.

In line 123 it is stated that leakage of fluid, proteins and lipids leads to swelling of Müller cells. I think leakage and swelling are two different phenomena that coexist during DME, but that there is no proof that one causes the other. Can the authors rephrase this or provide more evidence for this statement?

Figure 3: explain in the legend the action of sVEGFR1 and how that leads to the preservation of the stalk cell phenotype.

Figure 4: Bevacizumab can bind 2 molecules of VEGF, one on each arm.

Line 515: “Leukocyte-induced inflammation has been shown to play a significant role in the development of DME”. This is only shown in animal models, which do not fully represent the human situation. For a critical review, please read and include PMID: 28724696.

Line 519: Reference 100 does not show any evidence of tight junctional changes, only difference in FITC-conjugated dextran permeability. The permeability can as well be transcellular.

Figure 5: I would suggest to put the lower part in Fig2 and omit it in this figure. Legend text can be more explanatory. PDGFβshould be PDGFB.

Line 725: Faricimab is not shown in Fig4.

The studies with faricimab should be discussed more objectively, as there are essential differences from previous trials in terms of treatment intervals and concentrations administered. The superiority of faricimab hasn’t been shown to be true clinically. And faricimab trials do not examine Ang-2 inhibition as monotherapy. It is evaluating combination inhibition of VEGF and Ang-2; as it was non-inferior to aflibercept, which does not bind Ang-2, it remains to be shown the benefit of Ang-2 in combination to VEGF inhibition. The fact that a higher molar dose of anti-VEGF is used in the faricimab dose in the trial, which might account for anatomic benefits, though limited, may indicate that ang-2 effects remain unproven.

For integrin inhibitors, include THR-687, which has shown insufficient evidence of efficacy on the key endpoints.

Author Response

Re: Manuscript ID cells-1759286 entitled " Current and Novel therapeutic Approaches for Treatment of Diabetic Macular Edema"

*All co-authors have reviewed and agree with the following responses and revisions to the manuscript.

Reviewer 2

Comment 1: The review gives a nice overview of the current treatments for DME and includes beautiful illustrations. It is well written and it discusses the most important findings and new developments.

Response 1: Thank you very much for taking the time to review our work and for your positive as well as thoughtful comments.

Comment 2: Abstract and Line298, Line 699: the abbreviation for vascular endothelial growth factor is spelled wrong.

Response 2: Thank you catching these incorrect abbreviations. We have made corrections to abstract and line numbers 298 and 699.

Comment 3: Abstract: angiogenesis and neovascularization are mentioned in the same sentence, but have the same meaning.

Response 3: We have removed neovascularization this sentence (Line 24).

Comment 4: Abstract: need, perspective or novelty of this review should be mentioned to increase its value. The part that is at the end of the Introduction now.

Response 4: Thank you for this suggestion. We have added in two sentences at the end of the abstract to provide perspective (Lines 29-31).

Comment 5: Introduction: Type 2 diabetes mellitus was previously called non-insulin dependent diabetes mellitus (NIDDM) and late onset diabetes mellitus. These names are no longer used because they are inaccurate. Insulin is often used in the management of type 2 diabetes.

Response 5: We have corrected this sentence to remove old nomenclature and provided and additional sentence on the difference between the two (Lines 38-41).

Comment 6: Introduction: blindness caused by DM not only occurs in the US, also in other parts of the world. It will be appreciated if one looks a little further than just the US. Please add or replace reference.

Response 6: We have added in an additional sentence and reference detailing the impact of DM on a global scale as well as recent trends (Lines 44-46).

Comment 7: Introduction: Ref17 does not cover results of PKC-induced hypoxia and blood-retinal barrier (BRB) impairment. Please replace.

Response 7: We have added in three additional citations as references (Lines 94-96):

  • Williams, B., B. Gallacher, H. Patel and C. Orme (1997). "Glucose-induced protein kinase C activation regulates vascular permeability factor mRNA expression and peptide production by human vascular smooth muscle cells in vitro." Diabetes 46(9): 1497-1503.
  • Qaum, T., Q. Xu, A. M. Joussen, M. W. Clemens, W. Qin, K. Miyamoto, H. Hassessian, S. J. Wiegand, J. Rudge, G. D. Yancopoulos and A. P. Adamis (2001). "VEGF-initiated blood-retinal barrier breakdown in early diabetes." Invest Ophthalmol Vis Sci 42(10): 2408-2413.
  • Campochiaro, P. A. and A. Akhlaq (2021). "Sustained suppression of VEGF for treatment of retinal/choroidal vascular diseases." Prog Retin Eye Res 83: 100921.

Comment 8: Figure 1B: if possible, increase font size.

Response 8: We have increased font size of Figure 1 Panel B.

Comment 9: In line 123 it is stated that leakage of fluid, proteins and lipids leads to swelling of Müller cells. I think leakage and swelling are two different phenomena that coexist during DME, but that there is no proof that one causes the other. Can the authors rephrase this or provide more evidence for this statement?

Response 9: This is an excellent point. We agree. We believe that a more accurate term would be “concomitant”. Stepinac, Chamot, et. al (2005) associated vascular leakage with cellular edema of Müller cells and suggested that Müller cells can take up and accumulate fluid. However, glial swelling has also been suggested to precede the formation of extracellular edema (Yanoff, Fine, and Eagle (1984)). There is likely a contribution of both vascular and cytotoxic mechanisms that lead to DME. We have updated this sentence to better reflect our thought process (Lines 131-134).

Comment 10: Figure 3: explain in the legend the action of sVEGFR1 and how that leads to the preservation of the stalk cell phenotype.

Response 10: We have added details in the Figure 3 legend on the action of sVEGFR.

Comment 11: Figure 4: Bevacizumab can bind 2 molecules of VEGF, one on each arm.

Response 11: Thank you for catching this. We have updated Figure 4 to show 2 molecules bound.

Comment 12: Line 515: “Leukocyte-induced inflammation has been shown to play a significant role in the development of DME”. This is only shown in animal models, which do not fully represent the human situation. For a critical review, please read and include PMID: 28724696.

Response 12: Thank you for the point to this excellent reference. We have clarified in the sentence that this is data derived from animal models. In addition, we cited 28724696 and its opposing conclusions (Lines 525-528).

Comment 13: Line 519: Reference 100 does not show any evidence of tight junctional changes, only difference in FITC-conjugated dextran permeability. The permeability can as well be transcellular.

Response 13: Thank you for the suggestion. We have provided an additional reference on tight junction changes after administration of dexamethasone (Tian, Dong, et. al. (2007)).

Comment 14: Figure 5: I would suggest to put the lower part in Fig2 and omit it in this figure. Legend text can be more explanatory. PDGFβshould be PDGFB.

Response 14: We have removed the lower part in the Figure as suggested. PDGFβ has been changed to PDGFB. We have updated legend.

Comment 15: Line 725: Faricimab is not shown in Fig4.

Response 15: Thank you for catching this. We have updated Figure 4.

Comment 16: The studies with faricimab should be discussed more objectively, as there are essential differences from previous trials in terms of treatment intervals and concentrations administered. The superiority of faricimab hasn’t been shown to be true clinically. And faricimab trials do not examine Ang-2 inhibition as monotherapy. It is evaluating combination inhibition of VEGF and Ang-2; as it was non-inferior to aflibercept, which does not bind Ang-2, it remains to be shown the benefit of Ang-2 in combination to VEGF inhibition. The fact that a higher molar dose of anti-VEGF is used in the faricimab dose in the trial, which might account for anatomic benefits, though limited, may indicate that ang-2 effects remain unproven.

Response 16: These are great points. We have added in an additional paragraph at the end of this section discussing the trials more objectively as well as a mention of faricimab’s recent FDA approval (Lines 784-791).

Comment 17: For integrin inhibitors, include THR-687, which has shown insufficient evidence of efficacy on the key endpoints.

Response 17: We have added an additional paragraph discussing THR-687 (Lines 827-846).